# Environmental DNA gives comparable results to morphology-based indices of macroinvertebrates in a large-scale ecological assessment

**Jeanine Brantschen**[1,2]*, **Rosetta C. Blackman**[1,2,3], **Jean-Claude Walser**[4], **Florian Altermatt**[1,2,3]*

1 Department of Aquatic Ecology, Eawag, Swiss Federal Institute of Aquatic Science and Technology, Duebendorf, Zurich, Switzerland, 2 Faculty of Science, Department of Evolutionary Biology and Environmental Studies, University of Zurich, Zurich, Switzerland, 3 Research Priority Programme Global Change and Biodiversity (URPP GCB), University of Zurich, Zurich, Switzerland, 4 Department of Environmental Systems Science, Genetic Diversity Center, Federal Institute of Technology, Zurich, Switzerland

* Jeanine.Brantschen@eawag.ch (JB); Florian.Altermatt@ieu.uzh.ch (FA)

**Data Availability Statement:** All raw Illumina sequencing data files are available from the European Nucletoide Archive (project number PRJEB44539).

## Abstract

Anthropogenic activities are changing the state of ecosystems worldwide, affecting community composition and often resulting in loss of biodiversity. Rivers are among the most impacted ecosystems. Recording their current state with regular biomonitoring is important to assess the future trajectory of biodiversity. Traditional monitoring methods for ecological assessments are costly and time-intensive. Here, we compared monitoring of macroinvertebrates based on environmental DNA (eDNA) sampling with monitoring based on traditional kick-net sampling to assess biodiversity patterns at 92 river sites covering all major Swiss river catchments. From the kick-net community data, a biotic index (IBCH) based on 145 indicator taxa had been established. The index was matched by the taxonomically annotated eDNA data by using a machine learning approach. Our comparison of diversity patterns only uses the zero-radius Operational Taxonomic Units assigned to the indicator taxa. Overall, we found a strong congruence between both methods for the assessment of the total indicator community composition (gamma diversity). However, when assessing biodiversity at the site level (alpha diversity), the methods were less consistent and gave complementary data on composition. Specifically, environmental DNA retrieved significantly fewer indicator taxa per site than the kick-net approach. Importantly, however, the subsequent ecological classification of rivers based on the detected indicators resulted in similar biotic index scores for the kick-net and the eDNA data that was classified using a random forest approach. The majority of the predictions (72%) from the random forest classification resulted in the same river status categories as the kick-net approach. Thus, environmental DNA validly detected indicator communities and, combined with machine learning, provided reliable classifications of the ecological state of rivers. Overall, while environmental DNA gives complementary data on the macroinvertebrate community composition compared to the kick-net approach, the subsequently calculated indices for the ecological classification of river sites are nevertheless directly comparable and consistent.

**Funding:** The funders had no role in study design, data collection and analysis, decision to publish, or preparation of the manuscript. Funding for the project (to FA) is provided by the Swiss National Science Foundation Grant No 31003A_173074 and the Swiss Federal Office for the Environment (BAFU/FOEN).

**Competing interests:** The authors have declared that no competing interests exist.

## Introduction

Human activities change natural habitats and thereby inherently affect biodiversity [1]. Freshwater ecosystems are among the most affected and are facing steep biodiversity declines due to anthropogenic pressures [2]. To quantify changes in diversity and community structure of natural communities, monitoring of species is essential [3]. Over decades, ecologists have built an understanding of the responsiveness of certain taxonomic groups to pressures based on assemblages of communities and changes therein [4]. The ecological knowledge of species allows for the interpretation biodiversity pattern to assess environmental pressures and impacts on ecosystems. Routine biomonitoring often classifies ecosystem states through biotic indices that succinctly summarize the information of species assemblages, allowing for comparison to reference states or systems [5,6].

The classification of ecological integrity based on biotic indices is generally focused on certain taxonomic groups. In freshwater ecosystems, the most commonly used are fish, macroinvertebrates, macrophytes, and diatoms [7,8]. For these groups, monitoring generally involves the capture, preservation, and morphological identification of specimens in the field or laboratory. This type of monitoring can be costly in terms of time and money, requiring expert taxonomic skills and often missing rare, small, or elusive species [9].

In the last decade, molecular approaches have proven effectivity for the for the assessment of distributions of individual species (species-specific approach) and community assemblages (metabarcoding approaches). Thereby, DNA extracted from an environmental sample (so-called environmental DNA, eDNA) provides information about the possible occurrence and distribution of species [10–13]. Monitoring based on eDNA sampling has been explored for different indicator taxa, including fish (e.g., [14,15]); macroinvertebrates (e.g., [16–18]); macrophytes (e.g., [19,20]) and diatoms (e.g., [21,22]). In several instances, eDNA sampling was shown to complement traditional approaches for the assessment of biological indicators and the ecological state of ecosystems [23–25]. Most comparisons of monitoring indicator taxa have strongly focused on reproducing diversity patterns observed by traditional approaches and have paid less attention to possible challenges and opportunities of eDNA-based metabarcoding [26,27].

As eDNA and kick-net sample fundamentally different units (DNA vs. specimen) [28], the processing of samples and the interpretation of the detection cannot be compared one-to-one. The community recovered from an eDNA sample strongly depends on hydrological conditions of the body of water (e.g., transport of DNA) and choices in the downstream processing (e.g., the barcoding regions, primer choice). Contrastingly, traditional monitoring employs morphologically identifiable indicator organisms, thereby often using a subset of species as a para- or even polyphyletic group (e.g., [29]). For example, the widely used organisms belonging to the "macroinvertebrates" are defined by their size and function, and not by their phylogenetic unity. Targeting the genetic material of these phylogenetically dispersed organisms therefore aims for the coverage of a large portion of metazoans. It is well known that eDNA-based metabarcoding using generic primers (e.g., targeting Cytochrome Oxidase I) often result in limited taxonomic resolution and extensive amplification of non-target groups [30] such as rotifers or other small eukaryotes [31].

Technical biases of the metabarcoding data (e.g., PCR bias, sequencing errors) restrict the comparability to traditional count data [32,33] and limit the implementation in frameworks of existing biotic indices. Novel approaches to fully exploit and interpret molecular data are thus in demand [34,35]. A promising approach in the era of big data, machine learning has emerged for the analysis of high-dimensional and complex data [36] and recent studies have demonstrated the use of these approaches, such as random forest model [37], also in an ecological

context (e.g., [38–41]). A fundamental difference is the unit of OTUs vs. species used to calculate biotic indices, as not all OTUs are assigned to species. Taxonomy-free approaches accounting for the genetic diversity recovered in sequencing data can inform about features of communities outside of the classical indicator species concept. Such approaches may overcome the limitations of trying to match traditional and novel monitoring methods in a one-to-one manner and help to explore the opportunities, but also differences, between both approaches.

Here, we used eDNA metabarcoding for the detection of macroinvertebrate indicator taxa in a large-scale ecological assessment and addressed to which degree diversity pattern and the respective ecological index shows convergence. Kick-net and eDNA samples collected within the biomonitoring program for Swiss surface waters were used i) to compare community richness and composition estimates based on morphological identification versus eDNA metabarcoding data using the same indicator taxa, and ii) to evaluate how a supervised machine learning approach can be used for the prediction of the ecological state of rivers when also incorporating data on taxonomic groups not considered by the traditional approaches. We specifically focus on highly replicated and representative samplings based on a nationwide monitoring scheme run by the Swiss Federal Agency for the Environment, in order to cover all major river systems in Switzerland and to directly provide stakeholder-relevant conclusions.

## Methods

### Sample collection

The Swiss Federal Office for the Environment (FOEN) carries out routine monitoring of freshwater quality in Switzerland ("Nationale Beobachtung Obergewässerqualität", hereafter: NAWA). The goal of NAWA is to gather long-term reference data of the ecological state of riverine systems. Approximately 100 sites distributed over the major catchments in Switzerland are monitored regularly. The sampling scheme involves physicochemical parameters and the assessment of biological indicators (fish, macroinvertebrates, macrophytes, and diatoms), whereby macroinvertebrates are sampled using a standardized kick-net approach with subsequent morphological identification of species (see also [42]). In 2019, along with standard kick-net sampling, water samples for eDNA analyses were also taken at 92 of these sampling sites and were analyzed amplifying a barcode region for macroinvertebrates (Fig 1). More details of the sampling sites can be found in the supplements (S1 Table in S1 File), also providing information about the scores, the predictions and the classification for each site.

In brief: eDNA samples were collected in four replicates before the kick-net sampling. For each sampling site, two filter replicates were taken per riverbank (right and left bank, respectively, total n per site = 4). In these river systems, communities are not expected to systematically differ between the left and right riverbanks. Water was filtered on site (500 mL per filter, 2L per site), using a disposable sterile 60 ml syringe and Sterivex filters with a 0.22 μm pore size (Merck Millipore, Merck KgaA, Darmstadt, Germany). A total volume of 2 L was filtered per sampling site. Sterivex filters were then sealed with Luer caps (Merck Millipore, Merck KgaA, Darmstadt, Germany), put in a labeled plastic bag and placed in a cool box for short-term storage. After transport to the lab, filters were stored in the fridge at –20˚C until further processing. After placing the eDNA samples in the cool box, kick-net samples were collected upstream of the eDNA sampling sites in order to sample undisturbed habitats and to account for downstream transport of DNA, following standard procedures for the sampling of river systems [43]. The traditional sampling of macroinvertebrate groups by contractors followed commonly used kick-net methods [44]. In brief: At each site, the benthic fauna of eight microhabitats were sampled with a kick-net, each microhabitat was sampled for 30 seconds by disturbing the substrate by foot. Coarse organic material, sediment and gravel, and non-target

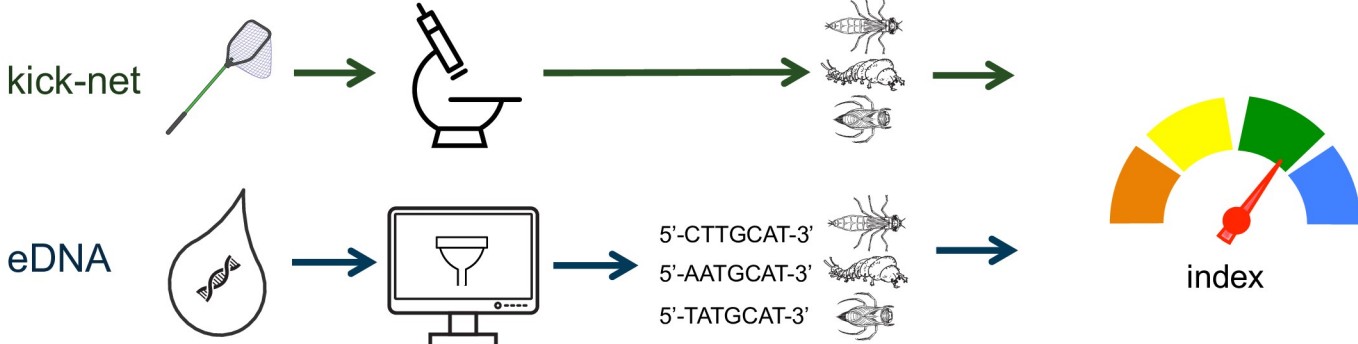

**Fig 1. Description of the study design.** A) The taxonomic composition of benthic macroinvertebrates at each sampling site was assessed with two methods: Kick-net and eDNA sampling. Subsequently, the focal index on the biological state (IBCH Index) was calculated from kick-net and eDNA data. B) Map of Switzerland showing the spatial setup of the biomonitoring sampling sites. Sampling sites are given as black points overlaid on the main network of rivers and lakes. Different blue shading highlights major catchments of Switzerland.

organisms such as fish or amphibians were removed from the sample on site. The remaining sample was preserved with 95% ETOH on site. In the laboratory, macroinvertebrate specimens were sorted and classified into 145 pre-defined taxonomic indicator groups (S2 Table in S1 File). These groups were mostly at the family level, except for Porifera, Bryozoa, and Cnidaria, which were only grouped at the phylum level.

## eDNA sample processing

The extraction of the eDNA samples took place in a specialized laboratory following clean lab procedures [45]. The DNA was extracted by using the Qiagen PowerWater Sterivex Extraction Kit (Qiagen, Germany) following the manufacturer's protocol. Extractions took place in batches of twelve samples. Field filter controls served as negative extraction controls and were extracted randomly among the samples. The extracted DNA was eluted in 100 μl elution buffer and stored at –20˚C until further processing.

## Library preparation

Samples were amplified using a 313 bp fragment of the COI marker. The primer pair used was mICOIintF and jgHCO2198 [46,47] with a modification to include the Nextera® transposase sequences (S3 Table in S1 File). All samples, negative and positive controls (the latter being a synthetic oligo, see S3 Table in S1 File) were randomized over four 96-well PCR plates. The first PCR was carried out in a total volume of 25 μL containing polymerase AmpliTaq Gold 360˚ (1.25 U/μL), 0.5 μM each of each primer, 1x Buffer I (Thermo Fisher Scientific, MD, USA), BSA (0.1 mg/μL), dNTP (0.2 mM), MgCl2 (1 mM), SigmaFree water and 2 μL of DNA template was added per reaction. The PCRs were performed on a thermal cycler (Biometra T1 Thermocycler, Analytik Jena GMBH, Ge) using the following touchdown protocol: initial denaturation at 95˚C for 10 min, the first 25 cycles started with the denaturation at 95˚C for 15 s, annealing at 62˚C for 30 s, followed by extension at 72˚C for 30 s. After this the cycler performed 16 cycles where the annealing temperature was reduced by one degree each cycle, performing the last cycle at a temperature of 45 degrees. Final extension was performed at 72˚C for 5 min before the plates were cooled down to 10˚C. All samples were tested for amplification success with the AM320 method on the QiAxcel Screening Cartridge (Qiagen, Germany). First step PCR products were cleaned with the ZR DNA Sequencing clean-up Kit (Zymo Research, USA) following the manufacturer's protocol with the minor modification by which the elution step was prolonged to 2 min. at 4000 g.

The clean amplicons were indexed using the Illumina Nextera XT Index Kit A and D following the manufacturer's protocol (Illumina, Inc., San Diego, CA, USA). A reaction contained 25 μL 2x KAPA HIFI HotStart ReadyMix (Kapa Biosystems, Inc., USA), 5 μL of each of the Nextera XT Index adaptors, and 15 μL of the DNA templates. The second reaction had the following PCR protocol: initial activation at 95˚C for 10 min, thermal cycling following a denaturation at 95˚C for 30 s; annealing at 55˚C for 30 s; extension 72˚C for 30 s. After 8 cycles of final extension at 72˚C for 5 minutes, they were cooled to 10˚C and stored in the fridge at 4˚C for downstream application. PCR products were then cleaned using the Thermo MG Magjet bead clean-up kit and a customized program for the KingFisher Flex Purification System (Thermo Fisher Scientific Inc., MA, USA) to remove excessive Nextera XT adaptors. The cleaned product was then eluted 50 μl in a fresh plate and stored at 4˚C.

## DNA quantification and normalization

Clean PCR products were quantified using the Qubit BR DNA Assay Kit (Life Technologies, Carlsbad, CA, USA). DNA of samples and the standard dilution series were quantified in

replicates on a Spark Multimode Microplate Reader (Tecan, US Inc., USA). Samples were united in a normalization step into equimolar pools, according to their respective concentration using the BRAND Liquid Handling Station (BRAND GMBH + CO KG, Wertheim, GE). Negative extraction and PCR controls were added according to their concentration. The final library was cleaned using SPRI beads (0.8 x) twice. Library concentration was quantified by the Qubit with the HS Assay Kit and amplicon size was verified on the Agilent 4200 TapeStation (Agilent Technologies, Inc., USA). The Nextera XT library prep Kit (Illumina, Inc. San Diego, CA, USA) was used before loading the library onto the flow cell with a 16 pM target concentration and 10% PhiX. Paired-end sequencing using v3 chemistry was performed on an Illumina MiSeq (Illumina, Inc. San Diego, CA, USA) at the Genetic Diversity Center (ETH, Zurich).

## Bioinformatic amplicon sequence analysis and quality filtering

The bioinformatics workflow for post-sequencing data processing used the following approach: the data quality of the demultiplexed reads was checked using FastQC [48]. Raw reads were first end-trimmed; merged and full-length primer sites removed using usearch (v11.0.667_i86linux64) [43]. The merged and primer trimmed reads were quality filtered using prinseq-lite (v0.20.4). The UNOISE3 (usearch v10.0.240) [44] method with an additional clustering at 99% identity was applied to obtain error corrected and chimera-filtered sequence variants (zero-radius OTUs) [45]. The invertebrate mt code was used to check for stop codons in sequences; retained were zOTUS (OTUs hereafter) with open reading frames. These OTUs were mapped against a customized COI reference for taxonomic assignments (S6 Information in S1 File). A more detailed workflow report file including parameter settings and documenting data loss are specified in the supplements (S7 Information in S1 File).

## Statistical analysis

All analyses were performed in R (v3.6.0) [49]. The data were imported using the package (v1.28.0) [50]. In the first step, raw data were filtered based on the detection of OTUs in negative and positive controls. For this, a read threshold of OTU detected in controls was calculated and subsequently subtracted from every sample. The threshold was based on the number of reads in controls versus the overall number of reads in all samples for this species, so for every OTU we calculated a minimum number of reads in a sample to count as a detection, lower read numbers were removed. Further, we checked the read depth of samples compared to controls. To reduce spurious and stochastically detected OTUs, the filter replicates per site were used to establish stringency filters. With the least stringent filter, an OTU had to be detected in 1 out of the 4 filters to be retained. Increasing stringency kept only OTUs detected in at least 2, 3, or 4 out of 4 filter replicates in the data. Here, we used the data with an OTU detection of 2 out of 4 filters. We tested the sensitivity of our results with respect to the two out of four threshold, and found that the results are qualitatively and quantitatively consistent with a less stringent threshold. To compare the detection of indicators from the kick-net and the eDNA data, a taxonomic filter removed non-target OTUs as defined by the monitoring framework (S2 Table in S1 File).

   After those filter steps, spatial patterns of indicator group richness based on presence-absence of indicators were plotted using Swissriverplot (v1.28.0) [51]. In a second step, the taxonomic composition was compared using ggalluvial (v0.12.0) [52] and ggplot2 (v3.3.2) [53]. For eDNA metabarcoding data, the proportion of a taxon was calculated as "reads of all OTUs within one indicator group/total number reads". For kick-net-samples, proportions are "counts within one indicator group/total number of counts".

   Under the framework of the Swiss-wide NAWA biomonitoring, the ecological state of rivers is evaluated based on the calculation of a biotic index from kick-net data [42]. The index

accounts for taxa diversity and taxa indicator values and is represented as a numerical score from 0 to 20. The higher the score, the less anthropogenic influence is recognizable at a site. The numerical scores are then categorized into five categories "bad" to "very good". For the calculation of the index, specimens captured in kick-net samples had been sorted, identified to phylum or family level (S2 Table in S1 File), and counted. The calculation is based on the diversity and indicator value of the taxa [44]. The eDNA data were also analyzed to predict the biotic index, using a supervised random forest model. In order to run the random forest model, we included only indicator macroinvertebrate groups, therefore we subset the eDNA data to the OTUs assigned to the following phyla: Arthropoda, Cnidaria, Porifera, Bryozoa, Mollusca. The presence-absence data of OTUs were used to train a random forest model based on a taxonomy-free approach [39]. A grid search was performed in caret (v6.0.86) [54] to establish the optimal value for the parameters mtry (n/3), node size (3), and ntrees (500). A random forest model with those optimal parameters was fitted for every sample using ranger (v0.12.1) [55]. Based on a random subset of the samples (mtry = n/3), the random forest classifier was trained using OTUs of indicator taxa as predictive features and the index score calculated from kick-net as the response variable. The classifier subsequently predicted the biotic index of a sample based on the OTU composition. This biotic index was predicted for every sample iteratively in ranger. The relationship between the observed and the predicted score was evaluated based on the adjusted $R^2$ of a linear model and the goodness of fit based on Cohen's Kappa $\kappa$ [56]. The level of agreement is described for all values as: $\kappa < 0.05$: no agreement, $0.05 < \kappa < 0.20$: very poor, $0.20 < \kappa < 0.40$: poor, $0.40 < \kappa < 0.55$: fair, $0.55 < \kappa < 0.70$: good, $0.70 < \kappa < 0.85$: very good, $0.85 < \kappa < 0.99$: excellent, and $\kappa = 1$: perfect.

## Results

### Summary of raw amplicon sequencing data

Next generation sequencing of eDNA generated 26.64 million reads of which 24.60 million passed the quality filter. After bioinformatic processing, 15.8 million reads were left for downstream analysis. The average sequencing depth per sampling site (4 filter replicates pooled) was 166,827 reads (range: 30,543–660,048), covering in total 7,231 OTUs. After removal of weak samples and cleaning of the raw data based on positive and negative controls, only OTUs detected in at least 2 out of 4 filter replicates per sampling site were retained in the data. This step decreased the mean read depth per sampling site to 147,913 reads (range: 40,891–602,621; Fig 2A) and the mean number of OTUs per sampling site to 835 (range: 118–1698; Fig 2B). In total 4,599 OTUs were retained after this "2 out of 4" filter step, and of those, 205 OTUs were assigned to the 145 taxonomic levels of indicator groups used for the calculation of the biotic index. The read abundance distribution indicates that the indicator OTUs generally have a relatively high read coverage but were also interspersed by non-target OTUs (Fig 2C). In the kick-net sampling, a total of 145 possible indicator taxa were assessed (S2 Table in S1 File), of which 98 were detected in this monitoring campaign.

### Local diversity pattern (alpha diversity)

To compare the alpha diversity pattern of macroinvertebrates from kick-net with eDNA sampling, indicator group richness was mapped for all 92 sampling sites in Switzerland (Fig 3). The observed mean richness at a site derived from kick-net was 23 indicator taxa (range: 11–41). Constraining the sequencing data to OTUs assigned to all possible indicator taxa, the eDNA approach led to the detection of 205 OTUs in 21 families (OTUs assigned to family, genus or species levels). A large portion of OTUs was not considered for the downstream analysis, either belonging to non-indicator taxa or lacking taxonomic assignments. The observed

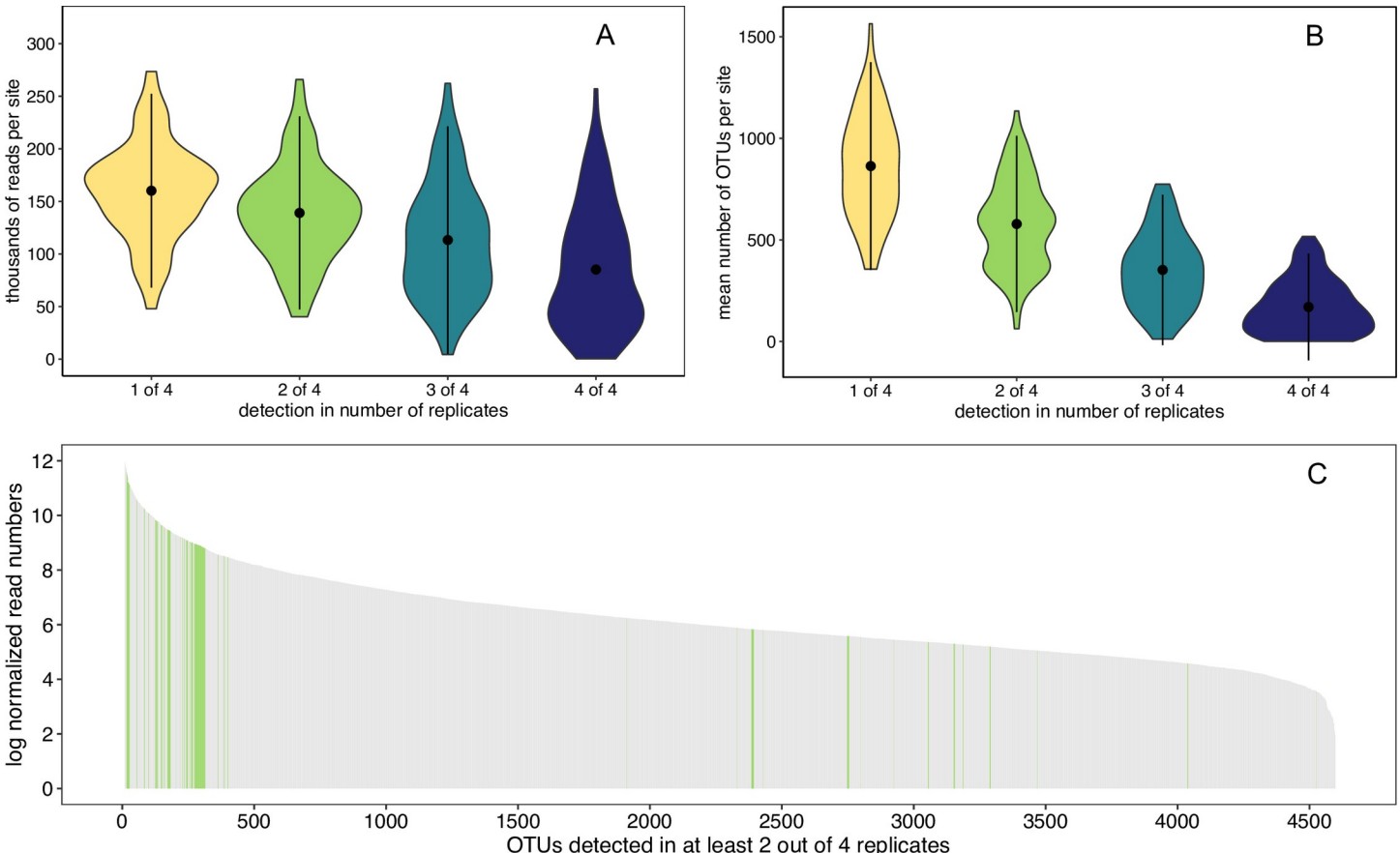

**Fig 2. Filtering of raw sequencing data.** Distribution of A) mean read number per sampling site and B) mean OTU number per sampling site using thresholds based on detection rate in the four field filter replicates. In the violin plots, the black dots indicate A) the mean read number over all sampling sites and B) the mean number of OTUs over all sites, the black vertical lines span the 95%-quantiles of all values. The detection rate is given with increasing stringency: Detection of an OTU in at least 1, 2, 3, or 4 out of 4 filter replicates per site, respectively. C) Read abundance distribution of OTUs (n = 4599) detected in at least 2 out of 4 replicates per site. Read abundances of OTUs that were taxonomically assigned to indicator taxa are highlighted in green.

mean richness of indicators by eDNA sampling was 9 indicator taxa (range: 2–18, Fig 3). A linear model showed little agreement in local richness pattern (adj. $R^2$ = 0.026, p = 0.08) detected by the two methods (S5 Fig in S1 File). However, eDNA detected significantly fewer indicator taxa at a site level (p < 0.001).

## Overall diversity pattern (gamma diversity)

The overall composition of indicator taxa (gamma diversity) detected by the two methods corresponded adequately for common groups (i.e., Diptera, Ephemeroptera, Plecoptera, Amphipoda). Abundances of indicators were rendered comparable by using read and count data on a log-transformed proportional scale (Fig 4). Abundant indicator taxa in the kick-net sampling were also identified as common groups derived from eDNA. The most frequently detected indicator taxa were Diptera, Ephemeroptera, Plecoptera, Trichoptera, and Amphipoda. Of those, Diptera and Ephemeroptera ranked equally in kick-net and eDNA data. Proportionally, Trichoptera was more abundant than Plecoptera in kick-net samples, but this ranking was reversed in eDNA data. For less frequently observed groups in the community, their relative rank varied between the two methods (e.g., Cnidaria, Isopoda, Coleoptera, Oligochaeta, Mollusca, or Gastropoda) (S4 Table in S1 File).

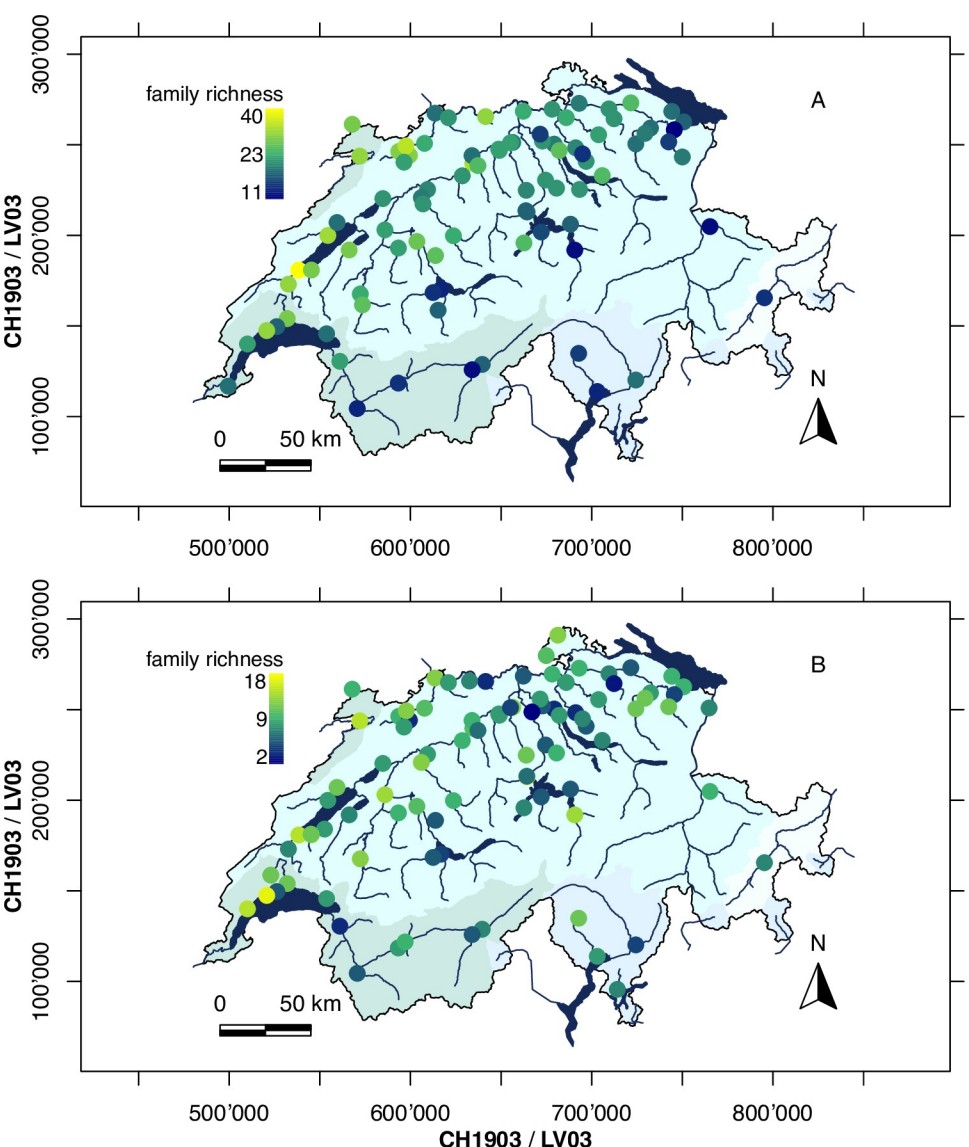

**Fig 3. Spatial richness pattern.** The taxonomic richness of indicator groups in Swiss rivers at each sampling site based on A) kick-net monitoring and B) eDNA monitoring. For the latter, only macroinvertebrates also considered in the traditional biological assessment are included. The color gradient is adjusted to the respective range of indicator richness values.

### Inference of the biotic index from eDNA data

The biotic index is calculated based on two components: one is the taxa richness, the other one is the occurrence of indicators groups. Kick-net and eDNA sampling methods picked up diverging richness patterns of indicator taxa, but largely showed a similar composition of indicator communities. The realized biotic index score was on average 13.3 (range: 8–17). The random forest model predictions were highly correlated with the biotic index scores observed from kick-net sampling ($R^2$ = 0.61, p < 0.001) (Fig 5).

Similar to the kick-net assessment, most predictions of the index were centered on the categories "intermediate" to "good", and only a few were predicted to fall into the categories "bad" or "very good" (Fig 6A). The majority of the predictions (72%) classified the ecological state of

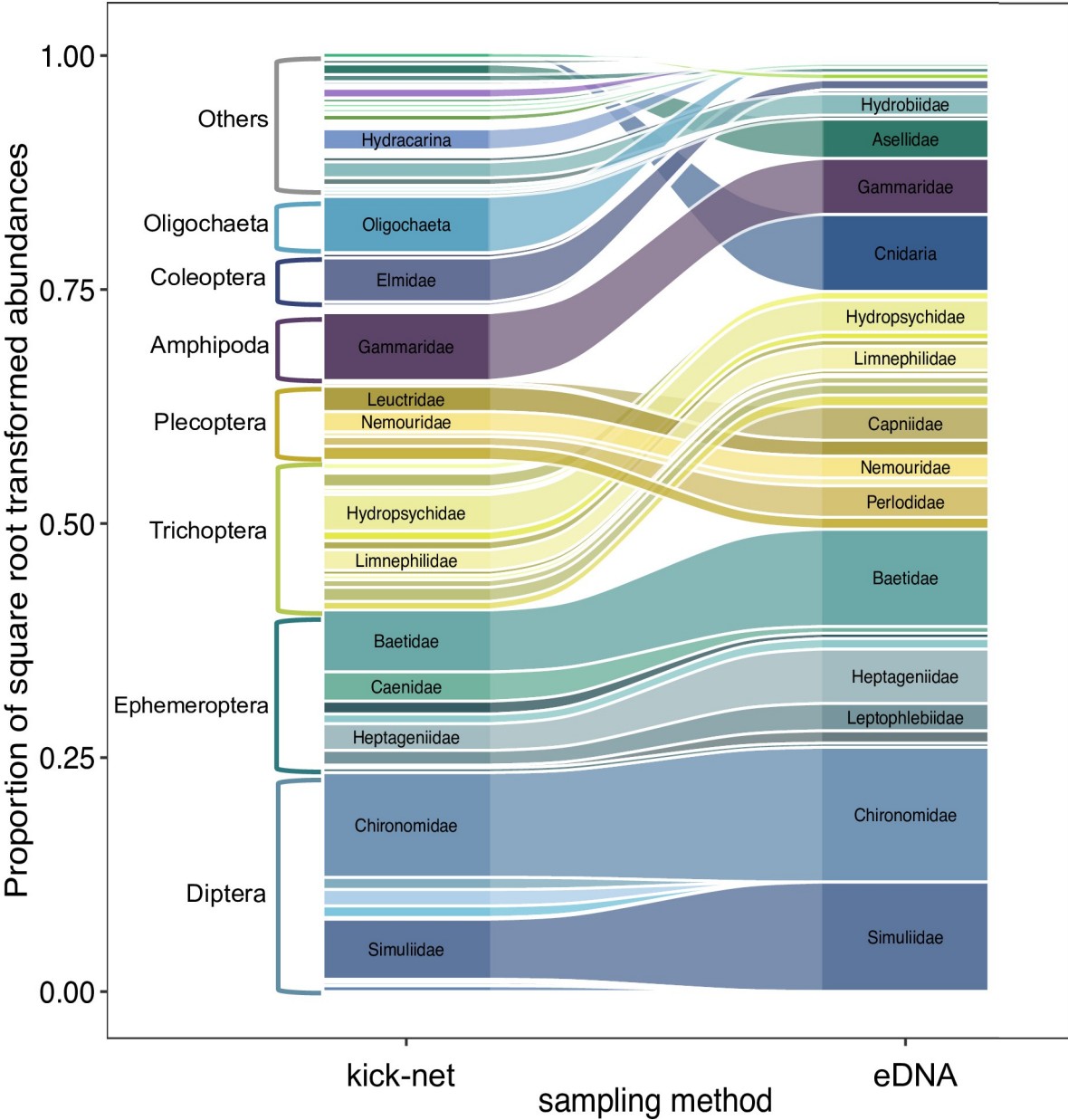

**Fig 4. Overall diversity pattern.** Proportions of indicator groups detected by kick-net versus eDNA monitoring. The stacked bars indicate proportions of indicator groups inferred by the two methods (proportion of counts for kick-net and of reads for eDNA). For the most common indicator groups, names are given. The flows between the two stacked bars connect families within indicator groups between the methods. A change in flow width indicates a change in proportion depending on the method used.

sampling sites correspondingly to the traditional kick-net-based estimates, and maximally diverged by one category (Fig 6B).

## Discussion

### Molecular surveys for traditionally established bioindicators

Despite the demonstrated suitability of eDNA for the survey of species and communities in aquatic ecosystems [10–13], routine implementation of molecular approaches in

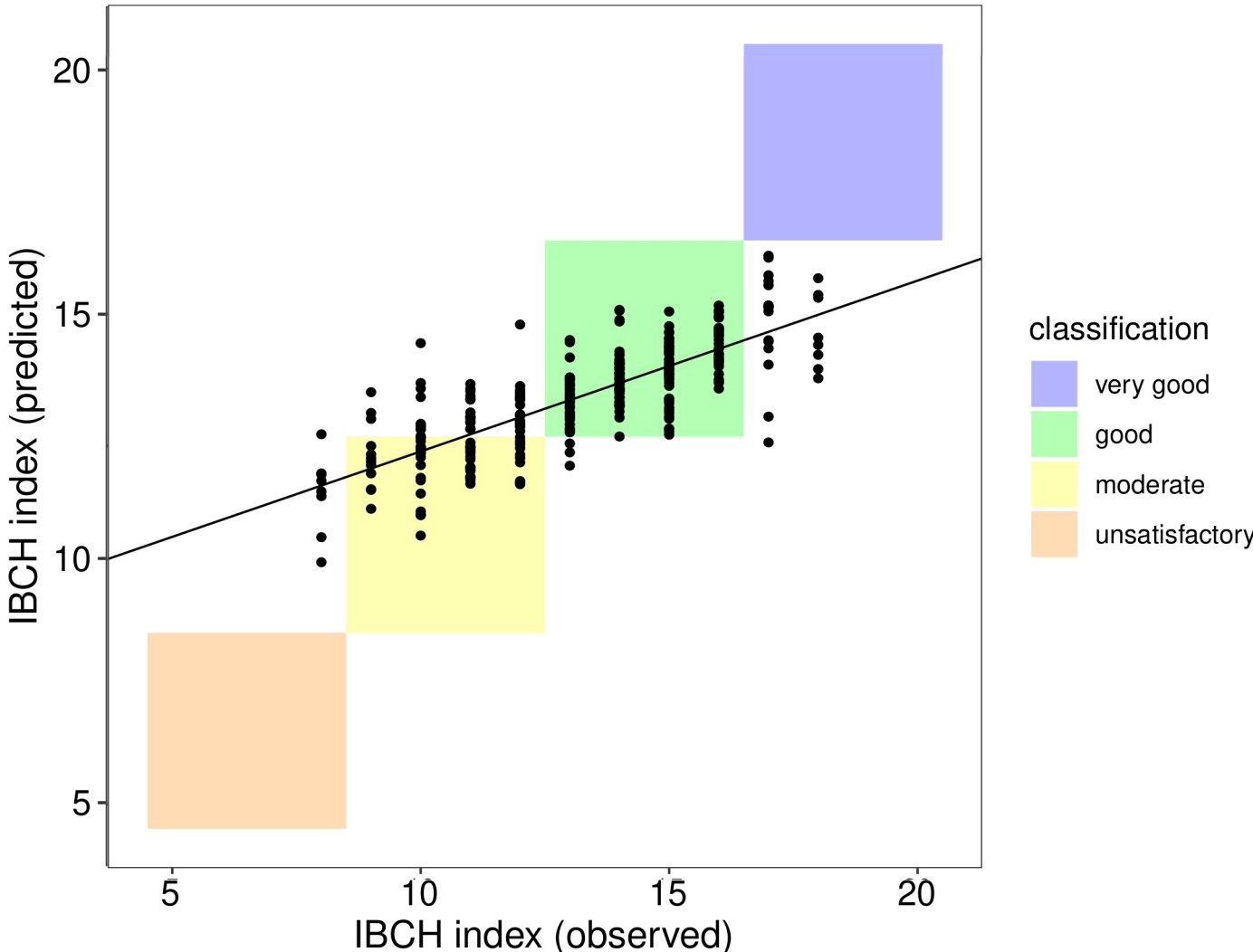

**Fig 5. Biotic index based on bioindicators.** Comparison of the index on the biological state (IBCH index) based on kick-net-derived scores (IBCH index observed) versus the predicted index derived from eDNA data. The predictions are the output from a random forest model deriving IBCH index scores using OTU presence-absence as input. A linear regression model gives the relationship between observed and predicted values (adjusted $R^2 = 0.61$, p-value $< 0.001$). The colored boxes summarize the numerical index scores ranging from 5 to 20 into categories ranging from "unsatisfactory" to "very good".

biomonitoring such as water quality assessments remains scarce. Here, we demonstrated the utility of water eDNA sampling for the assessment of macroinvertebrate-based ecological indicators on a national scale with a comparison to the traditional kick-net approach. Overall, the most common indicator taxa detected by eDNA were equally well covered by kick-net sampling at the national level (gamma diversity). However, local richness patterns (alpha diversity) were less consistent between the two methods, as eDNA samples on average detected fewer indicator taxa at the site level. Nevertheless, the composition of indicator OTUs at a site effectively informed about the ecological state, when using a machine learning algorithm. This study shows that eDNA is a valuable resource for the detection of macroinvertebrate indicator taxa and subsequent calculation of biotic indices, and it can be implemented for the ecological assessment of rivers.

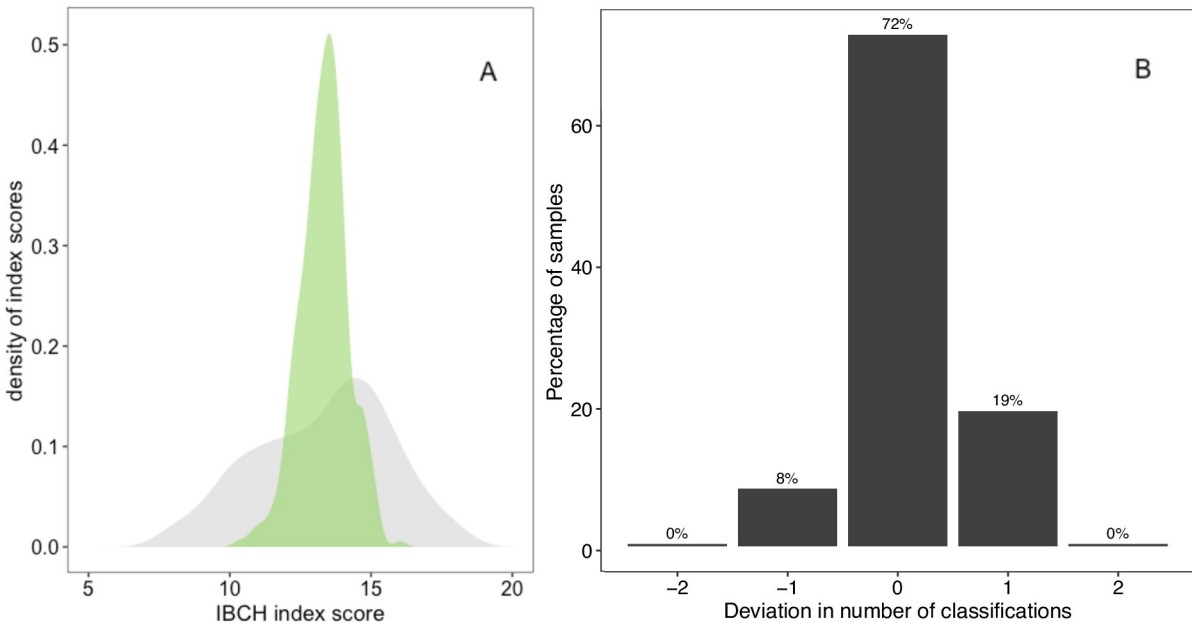

**Fig 6. Distribution of predicted classifications of the biotic state.** Comparison of the biological state of sampling sites when comparing classifications based on kick-net or random forest predictions. A) The density distributions for the observed (kick-net-based, grey) and the predicted (eDNA-based, green) IBCH index scores. The x-axis indicates the range of the biological index from 5 to 20. B) Barplot showing the percentage of sites that fell in the same (x = 0) or different (x ≠ 0) category by the random forest predictions based on eDNA data compared to the kick-net-based classifications. The majority of sampling sites were classified in the same category (72%). All other sites (28%) were maximally deviating by one category.

## Local and overall diversity patterns

Diversity patterns of macroinvertebrate communities in this study were restricted to the indicator groups as defined by the 145 taxonomic levels in the biotic IBCH index only. Significantly fewer indicator taxa were detected by eDNA sampling on a site level, although the ranking of the indicator taxa based on their overall relative abundance (proportions) in the community was similar for the most common groups. Diversity assessments of macroinvertebrate communities through COI metabarcoding have often reported similar or higher numbers of taxa compared to traditional methods (e.g., [17,18,57]), however, only a proportion of all reads are assigned to macroinvertebrates and even less to indicator taxa [58,59]. Arthropods are essential to the biotic assessment based on macroinvertebrates, and were the most dominant group among the OTUs assigned to bioindicators. These were consistently detected by eDNA and kick-net, with Diptera as the most common order, followed by orders with high indicator values, namely Ephemeroptera, Plecoptera, Trichoptera. However, some target groups were starkly underrepresented by eDNA, namely Hemiptera, Arachnida, and Coleoptera. The ecology of the target species delivers a possible explanation, e.g., their hydrophobic exoskeletons and the lower DNA shedding rates of these organisms [60], impairing these species' detection [61].

In river systems, DNA fragments are transported with the water flow [62,63], potentially mixing signals of locally occurring species as well as species only occurring further upstream. Thus, the two methods differ in the kick-net method being a truly local assessment, while the eDNA approach also integrating information on the communities along the stream [64]. The detection probability of target taxa can thus be influenced by the sampling strategy [65,66], and should generally consider the dendritic network structure of rivers [67]. This difference

may be explaining the partial mismatches of local richness or identity of organisms measured with the eDNA method compared to the kick-net approach, and result in more complementary than directly comparable diversity and composition estimates.

Another factor affecting the comparison between molecular and traditional methods is the taxonomic range of bioindicators used. The macroinvertebrate indicator groups used in the calculation of the biotic index are selected due to their known biotic responses to stressors and their size and are not a monophyletic group. In order to target those metazoans traditionally surveyed, the metabarcoding approach relies on highly degenerate primers [68], which amplify a wide range of non-target DNA fragments from eDNA samples. This approach has the drawback of unspecific amplification of non-target organisms at the cost of target organism sequences [30]. Therefore, for the comparison of diversity with the traditional approach, only a small fraction of the eDNA reads corresponding to macroinvertebrate indicator taxa are used in the biotic index. In combination with the high diversity detected with degenerate primers, this might hamper the detection of less abundant taxa at a site [16], as many reads are assigned to non-target taxa. As a possible solution, local richness measures may be improved by using more recently developed group-specific primers (e.g., [69]) or a combination of multiple markers for multiple groups [70]. With this, the eDNA metabarcoding would be more targeted towards the classical macroinvertebrate indicator taxa without the drawback of amplifying non-target taxa and lost read depth. Alternatively, the taxa considered could be extended and also include the many invertebrates amplified but not considered in the kick-net-based indices.

## Inference of the biotic index from eDNA data

Although eDNA unraveled lower indicator taxa richness locally, the overall community composition was similar for both methods and is thus promising for the implementation of eDNA-based biotic indices for water quality assessments. However, reads from a metabarcoding approach cannot be translated into species counts [32,33], and thus we could not directly calculate the IBCH index based on specimen counts for eDNA reads. As molecular monitoring provides different data than traditional surveys, novel approaches to fully exploit the eDNA derived community assemblages are needed [34,35]. Here, a supervised machine learning algorithm, i.e., random forest [71], was used to predict the biotic index based on the composition of indicator OTUs as predictive features of the ecological state of a site. This data-driven approach allows for the inclusion of the comprehensive list of indicator OTUs, as in contrast to the one-to-one filtering for the diversity measures, it is not based on taxonomically assigned OTUs as features. Instead of restricting the input data to the taxonomic levels of the biotic index (n = 205 OTUs), the machine learning algorithm included all OTUs belonging to surveyed macroinvertebrate phyla (n = 693 OTUs). That is, the machine learning approach was based on the OTU-level diversity of aquatic communities at a site. By this, we were able to include multiple OTUs previously united at family level, thus describe the community on a more nuanced level and therefore extract information not considered by the traditional index calculation. In the kick-net-based index calculations, only organisms captured by kick-net sampling, identified to family level at best and known to be responsive to stressors are included, whereas the eDNA survey is not restricted to these groups.

The incorporation of the OTU-level information resulted in highly comparable predictions of ecological status of the individual river sites, despite the previously described mismatch of local richness patterns. It showed that the great majority of predictions on the state of the river at a site corresponded with the classic approach, and maximally one category divergence between these two methods was observed. Importantly, a mismatch between the two methods

does not necessarily mean that the eDNA-based approach is less accurate, as both approaches are proxies (each with their inherent error) of a true state to be estimated.

The random forest model is well-suited to deal with high dimensional data [37] such as the community composition (observed OTUs) at each site. Using the kick-net-based observed ecological state as the response variable, the model is trained on a random subset of sites. After the training phase, it is then applied in order to infer the ecological state of any site based on the composition community, without the need of pre-assigning indicator values to the OTUs. This opens up the opportunity to use the information of metabarcoding more comprehensively and to shift away from the limited range of previously established indicator taxa. Machine learning as a data-driven approach could thus be used to identify sensitive taxa that were, due to limitations of the traditional methods, previously out of scope for ecological assessments. However, the predictive power of data-driven approaches is limited by the range of data. In this study, the values of the biotic index for most study sites were centered on the categories of "intermediate" and "good"; fewer observations were available to train models for moderate or very good sites, while none of the sites were classified as "poor". With a better coverage of all the possible categories of the index, the predictive power of the machine-learning algorithm could increase, as the distribution of averaged prediction from regression trees is narrower than the observed range of values [72]. Furthermore, the inference of causal links with random forest is limited. The data-driven predictions and the resulting classifications do not provide a mechanistic understanding of the biotic index calculation. The interpretation of misclassifications relies on the ecological understanding of the importance of input features, i.e. OTUs for the prediction. Despite these limitations, machine learning approaches have shown similar applicability for other biotic indicator taxa such as diatoms [24], macroinvertebrates [73], and taxonomy-free based approaches on prokaryotic and eukaryotic communities [39]. Overall, supervised machine learning can offer complementing or novel insights for the interpretation of big data in an ecological context, especially in the context of biotic indices when the like-for-like comparison is hindered by methodological differences.

## Conclusion

By carrying out a comparison of eDNA sampling with kick-net samples on a large scale, we take a crucial step in advancing the use of molecular methods for direct application in the assessment of the ecological state in routine monitoring programs. This study shows that eDNA sampling from water compared to kick-net data can give different estimates for the composition and diversity of macroinvertebrate communities at a local scale. Importantly, however, when assembling the community data on the level of the biotic states of river systems, both, kick-net and eDNA data indicate very comparable classifications of the ecological state. Thus, while the two methods are complementary at the level of biodiversity estimates, they still give comparable results at the level of ecological indices. Especially for biomonitorings, where the data on community composition and diversity is summarized into a biological index, eDNA can thus be a valid method to recover comparable assessments of ecological integrity.

## Supporting information

**S1 File. Additional information (S1-S7) supporting the study entitled "environmental DNA gives comparable results to morphology-based indices of macroinvertebrates in a large-scale ecological assessment".**
(DOCX)

## Acknowledgments

We thank Silvia Kobel and Aria Minder for technical advice in the laboratory. We thank the Swiss Federal Office for the Environment (BAFU/FOEN) and all the contractors for logistic support and the provision of the eDNA samples. The data analyzed in this paper were generated in collaboration with the Genetic Diversity Centre (GDC), ETH Zurich. We thank Noriko Uchida and the second, anonymous reviewer for their constructive feedback on our manuscript.

## Author Contributions

**Conceptualization:** Jeanine Brantschen, Florian Altermatt.

**Data curation:** Jeanine Brantschen, Jean-Claude Walser.

**Formal analysis:** Jeanine Brantschen, Jean-Claude Walser, Florian Altermatt.

**Funding acquisition:** Florian Altermatt.

**Investigation:** Jeanine Brantschen, Florian Altermatt.

**Methodology:** Jeanine Brantschen.

**Project administration:** Florian Altermatt.

**Resources:** Florian Altermatt.

**Supervision:** Rosetta C. Blackman, Florian Altermatt.

**Visualization:** Jeanine Brantschen, Florian Altermatt.

**Writing – original draft:** Jeanine Brantschen, Florian Altermatt.

**Writing – review & editing:** Jeanine Brantschen, Rosetta C. Blackman, Jean-Claude Walser, Florian Altermatt.

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
