## [Decision Letter · Decision Letter 0]

27 Jul 2021

PONE-D-21-15608

Environmental DNA is comparable to morphology-based indices of macroinvertebrates in a large-scale ecological assessment

PLOS ONE

Dear Dr. Brantschen,

Thank you for submitting your manuscript to PLOS ONE. After careful consideration, we feel that it has merit but does not fully meet PLOS ONE’s publication criteria as it currently stands. Therefore, we invite you to submit a revised version of the manuscript that addresses the points raised during the review process.

I got the recommendations and comments from two expert reviewers on the field. The both reviewer agree that the manuscript is technically sound and the data support the conclusions. However, lack of  home message were suggested by the reviewer and many of minor points, and I totally share their comments. Therefore, I can invite you to submit a revised version of the manuscript that addresses the points raised by the reviewers.

We look forward to receiving your revised manuscript.

Kind regards,

Hideyuki Doi

Academic Editor

PLOS ONE

“We thank Silvia Kobel and Aria Minder for technical advice in the laboratory. We thank the Swiss Federal Office for the Environment (BAFU/FOEN) and all the contractors for logistic support and the provision of the eDNA samples. The data analyzed in this paper were generated in collaboration with the Genetic Diversity Centre (GDC), ETH Zurich. Funding for the project (to FA) is from the Swiss National Science Foundation Grant No 31003A_173074 and the Swiss Federal Office for the Environment (BAFU/FOEN).”

Additional Editor Comments (if provided):

I got the recommendations and comments from two expert reviewers on the field. The both reviewer agree that the manuscript is technically sound and the data support the conclusions. However, lack of home message were suggested by the reviewer and many of minor points, and I totally share their comments. Therefore, I can invite you to submit a revised version of the manuscript that addresses the points raised by the reviewers.

Reviewers' comments:

Reviewer's Responses to Questions

**Comments to the Author**

1. Is the manuscript technically sound, and do the data support the conclusions?

Reviewer #1: Yes

Reviewer #2: Yes

2. Has the statistical analysis been performed appropriately and rigorously? 

Reviewer #1: Yes

Reviewer #2: Yes

3. Have the authors made all data underlying the findings in their manuscript fully available?

Reviewer #1: Yes

Reviewer #2: Yes

4. Is the manuscript presented in an intelligible fashion and written in standard English?

Reviewer #1: Yes

Reviewer #2: Yes

5. Review Comments to the Author

Reviewer #1: I am pleased to provide this review of Manuscript “Environmental DNA is comparable to morphology-based indices of macroinvertebrates in a large-scale ecological assessment” by Brantschen et al. The manuscript describes that macroinvertebrate survey using eDNA and traditional kick-sampling methods, and describing the characteristics of each data set and evaluating the river health indices based on each data set, complemented by machine learning.

In particular, the manuscript concisely argues that both of eDNA and traditional methods are each a proxy of true ecosystem status.

The manuscript is well structured therefore main statements are clear. Experimental methods and results are objective. All figures are beautiful and easy to understand. I put several questions and comments based on interest, not criticism. Main suggestion is to describe the characteristics of the sites where IBCH index show a category divergence between eDNA and traditional methods. If your team consider the content regarding my comments worthwhile, please add them to the manuscript.

Reviewer #2: In the article “Environmental DNA is comparable to morphology-based indices of macroinvertebrates in a large-scale ecological assessment” the authors have introduced a machine learning approach in order to calculate water quality indices based on macroinvertebrates composition as bioindicators. I am pleased to see that the approach can be comparable to the traditionally used method. However, I found the take home message a bit diluted through the manuscript and would like to propose some changes that I hope help to improve the current version.

Abstract: I would work a little bit more in the abstract as like it is right now, I don´t think it emphasizes enough what has been done. For example, machine learning approach is introduced at the end of last paragraph and for me it seems like an additional step, more than part of the methods. It is also quite contradictory that it is stated first that eDNA found less indicator taxa but then the indices were congruent? It is a bit difficult to follow if you don´t read the manuscript.

Lines 74-75 This is a very good point, maybe more emphasis on this in the introduction section?

Line 93 Effectivity instead of effective.

Line 101 I miss references here giving example, most sounds a bit vague.

Line 105 Very true, is there a reference to include here or just a personal statement?

Line 132 The predefined taxonomic groups: are not easy to understand.

Line 161 A diagram or figure summarizing sampling details would be really helpful, it can be supplementary or part of the Figure 1.

Line 165 State here how the 2L were taken: were 2L sampling site and n=4, then 500mL per sample/filter?

Line 169 Briefly explain here why upstream, or use a reference.

Line 218 Quantified instead of measured?

Line 246 How was this done?

Lines 247-248 What do you mean by low amplification?

Lines 258-259 I am not in favor of using read counts

Line 303 Why do you use this primer set instead the one developed by Eltbrech?

Line 320 I wouldn´t call this a weak correlation at all.

Line 400-401 The comparison of abundances makes no sense for me.

Lines 419-420 I wouldn´t say this is the explanation, as you are finding less indicator groups when using eDNA.

Lines 432-433 Could you use those non-targeted? Are they relevant? Flag species for example?

Line 444 I am confused about read counts, why did you calculate alpha and gamma div using relative abundances of reads and then not for the indices? I am not sure those diversities are giving relevant information when calculating using eDNA.

Line 466 So, if the machine learning approach is not employed, are the indices comparable?

Line 496 Is it worthy to make both methods complementary? Your reasoning through the manuscript is that the eDNA approach can be comparable, then, why complementary?

I find contradictory stating that one to one comparison is impossible but then directly comparisons are made with the diversity measures.

6. PLOS authors have the option to publish the peer review history of their article (what does this mean?). If published, this will include your full peer review and any attached files.

Reviewer #1: **Yes: **Noriko Uchida

Reviewer #2: No

---

## [Author Response · Author response to Decision Letter 0]

1 Sep 2021

Jeanine Brantschen

Überlandstr. 133

Department of Aquatic Ecology

Eawag

jeanine.brantschen@eawag.ch

 27th of August 2021

Response letter for our manuscript

Dear Prof. Dr. Hide Doi 

Firstly, we would like to thank you for your consideration and positive evaluation of our manuscript entitled “Environmental DNA gives comparable results to morphology-based indices of macroinvertebrates in a large-scale ecological assessment ”. 

The two expert reviewers gave very constructive feedback. We carefully considered and implemented their comments and their comments increased the reproducibility and the effective communication of our scientific findings. 

Please find below a copy of every comment made by the reviewer and our detailed response, the line numbers refer to the revised manuscript. We further added the revised manuscript and a version with all changes highlighted in yellow. We are confident that our changes have strengthened the manuscript.

On behalf of all authors, I would like to thank you for your time and for considering our study for publication in PLOS ONE. We look forward to hearing your decision. 

Yours sincerely, 

Jeanine Brantschen, on behalf of all authors

 

PONE-D-21-15608

Environmental DNA is comparable to morphology-based indices of macroinvertebrates in a large-scale ecological assessment

PLOS ONE

Dear Dr. Brantschen,

Thank you for submitting your manuscript to PLOS ONE. After careful consideration, we feel that it has merit but does not fully meet PLOS ONE’s publication criteria as it currently stands. Therefore, we invite you to submit a revised version of the manuscript that addresses the points raised during the review process.

Response: Thank you for your positive and highly constructive evaluation, and the offer to submit a carefully revised version. We have addressed all your and the reviewer’s comments and are happy to herewith submit a fully revised version of the manuscript that has addressed and resolved all comments.

I got the recommendations and comments from two expert reviewers on the field. The both reviewer agree that the manuscript is technically sound and the data support the conclusions. However, lack of home messages were suggested by the reviewer and many of minor points, and I totally share their comments. Therefore, I can invite you to submit a revised version of the manuscript that addresses the points raised by the reviewers.

Response: We thank both reviewers for their careful and positive evaluation of our manuscript. We are especially happy to hear that both agree on the soundness and conclusions of our manuscript. We have integrated their helpful comments to increase especially reproducibility and clarity with respect to the generalization of our study.

Response: We uploaded all the required documents named: 'Response to Reviewers', 'Revised Manuscript with Track Changes', and 'Manuscript'. Additionally, we expanded the supporting information file, therefore added the files 'Revised Supporting Information with Track Changes' and the file ' Supporting Information'.

Response: We respectfully decided not to upload our protocols, as all methods are described in detail in the method sections. The raw data are accessible on the open access repository of the European Nucleotide Archive. Furthermore, we happily provide protocols upon request to the corresponding authors.

We look forward to receiving your revised manuscript.

Kind regards,

Hideyuki Doi

Academic Editor

PLOS ONE

Response: Thank you for pointing out the style requirements. We adjusted the format of the manuscript to meet the PLOS ONE’s template, we included information on the ‘Supporting information’ including the captions of the supplements after the reference section. Also, we edited the title page, namely the authors and their affiliations, and we removed the keywords and ORCID-IDs to match the journals’ format requirements. 

Response: We added the information necessary in the updated ‘Funding Information’ section further down to make sure the „Financial Disclosure“ matches the „Funding Information“.

“We thank Silvia Kobel and Aria Minder for technical advice in the laboratory. We thank the Swiss Federal Office for the Environment (BAFU/FOEN) and all the contractors for logistic support and the provision of the eDNA samples. The data analyzed in this paper were generated in collaboration with the Genetic Diversity Centre (GDC), ETH Zurich. Funding for the project (to FA) is from the Swiss National Science Foundation Grant No 31003A_173074 and the Swiss Federal Office for the Environment (BAFU/FOEN).”

Response: We appreciate the comment and the updating of the online submission form on our behalf. We removed all the funding information from the acknowledgments section. Here, we provide an updated version of the Funding Statement, which reads as follow: „The funders had no role in study design, data collection and analysis, decision to publish, or preparation of the manuscript. Funding for the project (to FA) is provided by the Swiss National Science Foundation Grant No 31003A_173074 and the Swiss Federal Office for the Environment (BAFU/FOEN).”

Additional Editor Comments (if provided):

I got the recommendations and comments from two expert reviewers on the field. The both reviewer agree that the manuscript is technically sound and the data support the conclusions. However, lack of home message were suggested by the reviewer and many of minor points, and I totally share their comments. Therefore, I can invite you to submit a revised version of the manuscript that addresses the points raised by the reviewers.

Response: Thank you for your positive evaluation of our manuscript and for the invitation for resubmission. We clarified and expanded the statements regarding the main take-home messages, and added also a statement about the general importance of our scientific findings in the abstract, the discussion and the conclusion section. We thank the editor and the reviewer for this comment; the alterations improved the communication of the relevance of our study. All further points were considered as detailed in the sections below.  

Reviewers' comments:

Reviewer's Responses to Questions

Comments to the Author

1. Is the manuscript technically sound, and do the data support the conclusions?

Reviewer #1: Yes

Reviewer #2: Yes

2. Has the statistical analysis been performed appropriately and rigorously? 

Reviewer #1: Yes

Reviewer #2: Yes

3. Have the authors made all data underlying the findings in their manuscript fully available?

Reviewer #1: Yes

Reviewer #2: Yes

4. Is the manuscript presented in an intelligible fashion and written in standard English?

Reviewer #1: Yes

Reviewer #2: Yes

 

5. Review Comments to the Author

Reviewer #1: I am pleased to provide this review of Manuscript “Environmental DNA is comparable to morphology-based indices of macroinvertebrates in a large-scale ecological assessment” by Brantschen et al. The manuscript describes that macroinvertebrate survey using eDNA and traditional kick-sampling methods, and describing the characteristics of each data set and evaluating the river health indices based on each data set, complemented by machine learning. In particular, the manuscript concisely argues that both of eDNA and traditional methods are each a proxy of true ecosystem status.

The manuscript is well structured therefore main statements are clear. Experimental methods and results are objective. All figures are beautiful and easy to understand. I put several questions and comments based on interest, not criticism. Main suggestion is to describe the characteristics of the sites where IBCH index show a category divergence between eDNA and traditional methods. If your team consider the content regarding my comments worthwhile, please add them to the manuscript.

Response: We would like to thank reviewer #1 for the positive feedback and the constructive comments on our manuscript. We are especially happy to hear that our figures and results are appreciated from a visual and scientific perspective. All suggestions given are very helpful, and we addressed them. Specifically, we implemented responses to the questions in our manuscript and provided the suggested characteristics about all sampled sites in form of a table to the supplementary information. We added the text „More details of the sampling sites can be found in the supplements (Supporting information S1 Table).”

Comment Line 162: Were these sampling sites where the community would be expected to be different on the right and left banks?

Response: In the sampled river systems, we do not expect the communities within a single site to differ between the right and left bank because these rivers and streams are found in landscapes that have very similar land uses on both sides. We added this information to line 147, saying: “In these river systems, communities are not expected to systematically differ between the left and right river banks.” 

 

Comment Line 295: How would you support that using two of four is a reasonable strategy?

Response: We thank the reviewer for this comment. Indeed, which and how using different thresholds for quality filtering of metabarcoding data is an ongoing discussion (see also Mächler et al 2021, Molecular Ecology). Often, a quality-filtering step includes the removal of singletons or the definition of a certain read threshold (see for example: Blackman et al., 2020, nature scientific reports; Leese et al., 2021, Environmental DNA). 

Here, we decided to make use of the true replication, and only include records that were detected in at least two independent filter replicates. This decision was based on preliminary analyses, and we added this information. However, based on your comment we repeated the analysis with a less stringent (1 of 4) threshold. The results are both qualitatively and quantitatively very consistent (see additional figure 1 below or in the submitted response document), and the comparison with the kick-net data fit well (adj. R2 for 1 out of 4 threshold: 0.56; adj. R2 for 2 out of 4 threshold: 0.61). This high consistency is a strong indication for the robustness of our approach, and justifies the strategy. 

We added a sentence to line 241 to include this: “We tested the sensitivity of our results with respect to the 2 out of 4 threshold, and found that the results are qualitatively and quantitatively consistent with a less stringent threshold.“

Additional Figure 1: Biotic index based on bioindicators. Comparison of the index on the biological state (IBCH index) based on kick-net-derived scores (IBCH index observed) versus the predicted index derived from eDNA data. Here, the eDNA data used the present-absence data of all OTUs present in at least 1 out of 4 replicate filters per site (as opposed to 2 out of 4 in the main manuscript). The linear regression model gives the relationship between observed and predicted values (adj. R2 = 0.5629, p-value < 0.001).

Comment Line 304: Fig 3. Describe what the vertical lines and dots in violin plots are.

Response: We have added the following text to the Figure legend in line 297: “In the violin plots, the black points give A) the mean read number of all sampling sites and B) the mean number of OTUs over all sites; the black vertical lines indicate the 95%-quantiles of all values.”

Comment Line 321: Would the results be the same if the taxa detected one of four was included? Since the replicates in this manuscript is not very robust (two from the right and two from the left bank), it is possible that the indicator taxa are only included in one sample if the right and left banks are heterogeneous environment.

Response: We appreciate the reviewer’s feedback on our replication. As said above, the river systems at hand are intermediate to large in size and anthropogenically impacted (homogenised), and both riverbank sides are generally highly similar. We now added the sentence on these systems having generally homogeneous left and right river bank sides. Thus, we expect that sampling both riverbanks and combining this data will result in a representative coverage of the community. However, based on your comment we also repeated the analysis, and included all data (i.e., also 1 out of 4 threshold). The results of this additional analysis for the Random Forest classification are very similar when using a threshold of 1 out of 4 (adj. R2 = 0.56 versus adj. R2 = 0.61. (See also response above).

Comment Line 367: What are the characteristics of sampling sites that mismatch the results of the ecological state evaluation of traditional sampling method?

Response: This is indeed an intriguing question. We addressed this comment by adding a table with site characteristics of all sites, highlighting the sites that had a mismatch in the random forest classification. There was, however, no obvious environmental parameters associated to those sites. Thus, we cannot say if these mismatches are simply sampling artifacts, temporal changes, or are linked to further unmeasured factors. We added the text „More details of the sampling sites can be found in the supplements (S1 Table), also providing information about the scores, the predictions and the classification for each site.” 

 

Reviewer #2: In the article “Environmental DNA is comparable to morphology-based indices of macroinvertebrates in a large-scale ecological assessment” the authors have introduced a machine learning approach in order to calculate water quality indices based on macroinvertebrates composition as bioindicators. I am pleased to see that the approach can be comparable to the traditionally used method. However, I found the take home message a bit diluted through the manuscript and would like to propose some changes that I hope help to improve the current version.

Abstract: I would work a little bit more in the abstract as like it is right now, I don´t think it emphasizes enough what has been done. For example, machine learning approach is introduced at the end of last paragraph and for me it seems like an additional step, more than part of the methods. It is also quite contradictory that it is stated first that eDNA found less indicator taxa but then the indices were congruent? It is a bit difficult to follow if you don´t read the manuscript.

Response: We appreciate the evaluation of reviewer #2 and would like to thank them for the constructive comments on our manuscript. We implemented changes to the structure of the abstract, in order to emphasize the use of the random forest model, and provided concise conclusions about the relevance of this our study for the field. We are especially thankful for your point about contradictory statements and strengthened the home messages, improving the readability and to put our study in a broader context. 

Lines 74-75 This is a very good point, maybe more emphasis on this in the introduction section?

Response: We appreciate the interest in our approach and we addressed this in the introduction in a section previously (lines 98-101), and we expanded on this point by adding the following sentences to the manuscript (lines 101-105): „A fundamental difference is the unit of OTUs vs. species used to calculate biotic indices, as not all OTUs are assigned to species. Taxonomy-free approaches using the genetic diversity covered by the sequencing could inform about important features of communities outside of the classical species concept”.

Line 93 Effectivity instead of effective.

Response: We thank the reviewer for this comment and we changed the sentence to „In the last decade, molecular approaches have proven effectivity for the…“ (line 69).

Line 101 I miss references here giving example, most sounds a bit vague.

Response: To support our statement, we added reference to the section (line 80). This statement is further supported by the examples for individual indicator taxa, in the lines 74-76. 

Line 105 Very true, is there a reference to include here or just a personal statement?

Response: We appreciate the feedback and referenced our statement (line 82). 

Line 132 The predefined taxonomic groups: are not easy to understand.

Response: We appreciate pointing out the wordy description and we changed the expression to „indicator taxa“ in line 133. 

Line 161 A diagram or figure summarizing sampling details would be really helpful, it can be supplementary or part of the Figure 1.

Response: In order to give more sampling details, we added more information about the sampling scheme in form of a table that can be found in the supplementary information S1 Table. 

Line 165 State here how the 2L were taken: were 2L sampling site and n=4, then 500mL per sample/filter?

Response: We happily clarified this point by adding the following information to the text: „(500 mL per filter, 2L per site)” in line 149. 

Line 169 Briefly explain here why upstream, or use a reference.

Response: We added information to the phrase as an explanation to line 155: „in order to sample undisturbed habitats and to account for downstream transport of DNA.”

Line 218 Quantified instead of measured?

Response: We changed the phrase to „measured“.

Line 246: How was this done?

Response: We added the following sentence to line 233 to clarify: „The threshold was based on the number of reads in controls versus the overall number of reads in all samples for this species, so for every OTU we calculated a minimum number of reads in a sample to count as a detection, lower read numbers were removed.”

Lines 247-248 What do you mean by low amplification?

Response: We clarified this sentence in line 236 by rephrasing to “Further, we checked the read depth of samples compared to the controls.”

Lines 258-259 I am not in favor of using read counts

Response: We agree that read counts need to be cautiously used. We are aware that there is n ongoing debate about how to use them, and it is not our goal to take a strong position in this discussion. We thus state this in the discussion section line 436-438: 

„However, reads from a metabarcoding approach cannot be translated into species counts [29,30], and thus we could not directly calculate the IBCH index based on specimen counts for eDNA reads.” 

Further, in the analysis of gamma diversity, we use proportions of reads as a proxy of the detectability of the indicators with the eDNA method and we do not use reads for individual species abundances. For the further analysis of alpha diversity of indicator groups and the calculation of the ecological indices, we used presence-absence data, especially as we also agree with you on the caution needed with read numbers from metabarcoding approaches.

Line 303 Why do you use this primer set instead the one developed by Eltbrech?

Response: The primers we use here target the same barcode region like the suggested primers by Elbrecht and were developed by Leray (2013) and Geller (2013). These primers are commonly used for eDNA metabarcoding for the description of macroinvertebrates (see for example Cahill et al., 2018, Ecology & Evolution; Harper et al., 2021, Molecular Ecology, Nguyen et al., 2020, nature scientific reports) and we have used them for previous projects, thus we built on this knowledge of similar data sets.

Line 320 I wouldn´t call this a weak correlation at all.

Response: We appreciate the comment and rephrased the sentence in line 313 to: “A linear model showed little agreement in local richness pattern (adj. R2 = 0.026, p = 0.08) detected by the two methods (S5 Figure).“

Line 400-401 The comparison of abundances makes no sense for me.

We agree that there is a debate on how quantitative read counts are when working with eDNA metabarcoding in order to reflect species abundances in the field. We also agree that this relationship is weak at best. Importantly, the comparison of alpha diversity and the index calculation is based on presence-absence data. We also put emphasis on this aspect of abundance-estimate debate in our discussion line 436-438 and support our statement with recent literature (Elbrecht & Leese, 2015, Plos One; Piñol et al., 2019, Molecular Ecology). In our analysis, we do not use reads to reflect abundances of organisms on a site level, but we use proportions of the sequencing data to reflect the dominance of indicator group in our data. 

Lines 419-420 I wouldn´t say this is the explanation, as you are finding less indicator groups when using eDNA.

Response: We thank the reviewer for this comment on our explanation. We clarified our statement by rephrasing this paragraph to the following: „as DNA is distributed homogenously in the water column and thus not all local organisms are detected.”

Lines 432-433 Could you use those non-targeted? Are they relevant? Flag species for example?

Response: We thank the reviewer for his interest in the our discussion. Indeed, the non-targeted OTUs could present features of communities that were neglected in the traditional monitoring. This extra information could be used to inform data-driven approaches, as discussed in the line 483-488. However, robust ecological understanding of the species and the systems is needed, to ground-truth the analysis. 

Line 444 I am confused about read counts, why did you calculate alpha and gamma div using relative abundances of reads and then not for the indices? I am not sure those diversities are giving relevant information when calculating using eDNA.

Response: We used presence/absence data for number of indicator families in the calculation of alpha diversities on a site level, and presence-absence data of OTUs assigned to indicators for the calculation of the ecological index. We ensured that this is clearly stated in the method section in the sentence line 246: “After those filter steps, spatial patterns of indicator group richness based on presence-absence of families were plotted using Swissriverplot (v1.28.0).” 

Line 466 So, if the machine learning approach is not employed, are the indices comparable?

Response: It would be indeed interesting to compare the directly calculated index to the random forest approach. However, the calculation of the biotic index directly from metabarcoding data is not possible, as reads do not scale with counts. This means the formula used for calculation from kick-net data categorizes counts into discrete classes (<3, 3-10, 10-100, >100). Those values are meaningless when applied to reads. This would mean that the scores calculated from reads would result in score that are not interpretable.

Line 496 Is it worthy to make both methods complementary? Your reasoning through the manuscript is that the eDNA approach can be comparable, then, why complementary? I find contradictory stating that one to one comparison is impossible but then directly comparisons are made with the diversity measures.

Response: In our results, we show that eDNA gives different measures of local diversity (Figure 3) when the taxa groups are restrained to the bioindicators. However, using a universal primer pair we gathered information of organisms not represented in the extant ecological assessment (so called „non-target taxa“). Therefore, we mentioned that eDNA can give complementary information to kick-net sampling on the level of biodiversity. However, when considering the results on the level of the biotic index, the eDNA indices where based on OTUs assigned to the indicator taxa. The random forest model gives comparable results to the kick-net based on family-level of those indicator groups. 

To clarify our conclusion from messages, we added the following text to the conclusion line 490-501: „By carrying out a comparison of eDNA sampling with kick-net samples on a large scale, we take a crucial step in advancing the use of molecular methods for direct application in the assessment of the ecological state in routine monitoring programs. This study shows that eDNA sampling from water compared to kick-net data can give different estimates for the composition and diversity of macroinvertebrate communities at a local scale. Importantly, however, when assembling the community data on the level of the biotic states of river systems, both, kick-net and eDNA data indicate very comparable classifications of the ecological state. Thus, while the two methods are complementary at the level of biodiversity estimates, they still give comparable results at the level of ecological indices. Especially for biomonitorings, where the data on community composition and diversity is summarised into a biological index, eDNA can thus be a valid method to recover comparable assessments of ecological integrity.“ 

Further comments: We changed the acknowledgments in order to thank the two reviewers for their effort and feedback improving our manuscript. We added the following sentence to the Acknowledgement section (line 508): „We thank Noriko Uchida and a second, anonymous, reviewer for their valuable comments and constructive feedback on our manuscript.“ 

6. PLOS authors have the option to publish the peer review history of their article (what does this mean?). If published, this will include your full peer review and any attached files.

Yes, we would be open to publish the review history of our manuscript. 

Do you want your identity to be public for this peer review? For information about this choice, including consent withdrawal, please see our Privacy Policy.

Reviewer #1: Yes: Noriko Uchida

Reviewer #2: No

Response: We uploaded all our 6 figures to PACE and each of them was successfully converted to a valid file (.TIF), meaning the figures correspond to the PLOS requirements.

---

## [Editor Report · Decision Letter 1]

3 Sep 2021

Environmental DNA gives comparable results to morphology-based indices of macroinvertebrates in a large-scale ecological assessment

PONE-D-21-15608R1

Dear Dr. Brantschen,

We’re pleased to inform you that your manuscript has been judged scientifically suitable for publication and will be formally accepted for publication once it meets all outstanding technical requirements.

Kind regards,

Hideyuki Doi

Academic Editor

PLOS ONE

Additional Editor Comments (optional):

I carefully checked the revised manuscript as well as the response letter. I agree the revisions according to the reviewers’ comments and now can recommend to publish the paper in this journal.
---

## [Editor Report · Acceptance letter]

10 Sep 2021

PONE-D-21-15608R1 

Environmental DNA gives comparable results to morphology-based indices of macroinvertebrates in a large-scale ecological assessment 

Dear Dr. Brantschen:

I'm pleased to inform you that your manuscript has been deemed suitable for publication in PLOS ONE. Congratulations! Your manuscript is now with our production department. 

Kind regards, 

on behalf of

Dr. Hideyuki Doi 

Academic Editor

PLOS ONE